# Dual-phase nanostructuring of layered metal oxides for high-performance aqueous rechargeable potassium ion microbatteries

Ying-Qi Li [1,2], Hang Shi [1,2], Sheng-Bo Wang [1,2], Yi-Tong Zhou [1], Zi Wen [1], Xing-You Lang [1] & Qing Jiang [1]

Aqueous rechargeable microbatteries are promising on-chip micropower sources for a wide variety of miniaturized electronics. However, their development is plagued by state-of-the-art electrode materials due to low capacity and poor rate capability. Here we show that layered potassium vanadium oxides, $K_xV_2O_5 \cdot nH_2O$, have an amorphous/crystalline dual-phase nanostructure to show genuine potential as high-performance anode materials of aqueous rechargeable potassium-ion microbatteries. The dual-phase nanostructured $K_xV_2O_5 \cdot nH_2O$ keeps large interlayer spacing while removing secondary-bound interlayer water to create sufficient channels and accommodation sites for hydrated potassium cations. This unique nanostructure facilitates accessibility/transport of guest hydrated potassium cations to significantly improve practical capacity and rate performance of the constituent $K_xV_2O_5 \cdot nH_2O$. The potassium-ion microbatteries with $K_xV_2O_5 \cdot nH_2O$ anode and $K_xMnO_2 \cdot nH_2O$ cathode constructed on interdigital-patterned nanoporous metal current microcollectors exhibit ultrahigh energy density of 103 mWh cm$^{-3}$ at electrical power comparable to carbon-based microsupercapacitors.

[1] Key Laboratory of Automobile Materials (Jilin University), Ministry of Education, and School of Materials Science and Engineering, Jilin University, 130022 Changchun, China. [2] These authors contributed equally: Ying-Qi Li, Hang Shi, Sheng-Bo Wang. Correspondence and requests for materials should be addressed to X.-Y.L. (email: xylang@jlu.edu.cn) or to Q.J. (email: jiangq@jlu.edu.cn)

Continuous miniaturization of self-powered flexible and portable electronics, implantable medical devices, and microelectromechanical systems (MEMS) has raised urgent demands for sufficiently compact and/or flexible energy storage to realize safe and reliable energy autonomy[1,2]. Many types of miniaturized energy-storage devices are being developed to power microelectronic devices as stand-alone micropower sources or intermediate energy-storage units complementing energy conversion devices (e.g., solar cells, piezoelectric nanogenerators, or thermalelectric cells)[1,2]. These include traditional lithium-ion microbatteries based on volumetric reactions[3–7] and microsupercapacitors based on surface adsorption/desorption or/ and redox reactions[8–17]. Although lithium-ion microbatteries have high gravimetric/volumetric energy[3–7], the safety issues associated with highly toxic and flammable organic electrolytes besides the inferior rate capability and stability essentially impede their practical use in microelectronic devices. Whereas microsupercapacitors can tackle these problems by making use of (pseudo)capacitive energy storage in low-cost and safe water-based electrolytes, few of them have energy density approaching that of microbatteries[8–17]. Therefore, it is highly desirable to explore alternative electrochemical energy-storage technologies that can store/deliver energy with lithium-ion microbattery-like capacity and microsupercapacitor-like rate performance for satisfying the versatile requirements in micro/nano-systems.

Battery chemistries based on electrochemical intercalation of alkaline or alkali-earth cations (M = Li[+] (ref. [18–20]), Na[+] (ref. [21–24]), K[+] (ref. [25–28]), Mg[2+] (ref. [29]), and Zn[2+] (refs. [30,31])) in aqueous electrolytes are currently the most encouraging tentative, wherein the hydrated metal cations, instead of their crystalline ions, serve as charge carriers to take part in the electrochemical processes because of strong ion-solvent interactions[32]. By virtue of the K[+] cations having the smallest hydrated radii and the highest conductivity in water-based electrolytes[33,34], aqueous rechargeable potassium ion microbatteries (AR-PIMBs) hold a great promise as highly safe energy-storage microdevices to store/deliver high-density energy at fast charge/discharge rates. However, few electrode materials such as transition-metal oxides (TMOs) with a layered structure (e.g., polymorphic vanadium oxides[25]) and Prussian Blue analogs (PBAs)[26,27] (nickel hexacyanoferrate[35], iron hexacyanoferrate dihydrate[49], and nickel ferrocyanide[36]) have been explored for reversible hydrated K[+] cation intercalation. Compared with these PBAs with limited capacities (~60 mAh g[−1]), layered vanadium pentoxides without/with pre-intercalation of alkaline or alkali-earth ions, typically amorphous $V_2O_5$ xerogel ($V_2O_5 \cdot nH_2O$)[37,38] and $M_xV_2O_5 \cdot nH_2O$[25,39], have large interlayer distances, which are expected to facilitate diffusion and storage of guest hydrated cations with a high theoretical capacity of 443 mAh g[−1]. Nevertheless, the state-of-the-art layered vanadium pentoxide-based electrode materials often exhibit practical capacities far below the theoretical value and deliver much lower levels of electrical power than capacitor materials[40,41]. This is due to the basic structure units comprised of bilayer $V_2O_5$ sheets and a large number of crystalline water molecules sandwiched between them[34], wherein the former suffers from a low electrical conductivity and the latter blocks the intercalation of hydrated cations[42,43]. When removing the crystalline water to alleviate the blocking influence on the intercalation of hydrate cations, crystallization usually takes place to produce the orthorhombic $V_2O_5$ (o-$V_2O_5$) or the monoclinic $M_xV_2O_5$ with small interlayer spacing[38,40,44,45], which not only fails to improve the specific capacity and rate capability but leads to extra massive volume change during the insertion/extraction of hydrated cations to limit microbattery lifetime[38–41,43]. Such structure tradeoff always leads to their low capacity, poor rate capability, and short lifetime, restricting the wide use of vanadium oxide-based electrode materials in AR-PIMBs.

Here we show layered potassium vanadium oxides composed of amorphous/crystalline $K_{0.25}V_2O_5 \cdot nH_2O$/o-$V_2O_5$ dual phases (ac-$K_xV_2O_5$) as promising anode materials of high-performance AR-PIMBs when integrated on three-dimensional (3D) bicontinuous nanoporous Au current microcollectors (NP Au/ac-$K_xV_2O_5$). Therein, the amorphous $K_{0.25}V_2O_5 \cdot nH_2O$ serves as molecular pillars to keep the large interlayer spacing while the crystalline o-$V_2O_5$ offers sufficient room to accommodate more guest hydrated K[+] cations. As a consequence, the NP Au/ac-$K_xV_2O_5$ electrodes exhibit a volumetric capacity of as high as ~715 mAh cm[−3] with an exceptional rate performance. This enlists the AR-PIMBs with the ac-$K_xV_2O_5$ anode and the cryptomelane $K_xMnO_2 \cdot nH_2O$ (c-$K_xMnO_2$) cathode integrated on interdigital-patterned nanoporous Au current microcollectors (V-Mn) to store/deliver charge with volumetric energy of ~103 mWh cm[−3] (14-fold higher than that of 4V-500 μAh thin-film lithium battery) at electrical power comparable to carbon-based microsupercapacitors. Meanwhile, these V-Mn AR-PMIBs exhibit a long-term cycling stability because the large interlayer spacing can effectively relieve volume expansion/contraction during the insertion/extraction of hydrated K[+]. The superior electrochemical properties make them promising candidates as micropower sources to complement other energy conversion devices such as commercial solar cells.

## Results

**Preparation and structural characterizations.** Our fabrication strategy of NP Au/ac-$K_xV_2O_5$ microelectrodes is to electrodeposit the hydrated amorphous $K_xV_2O_5 \cdot nH_2O$ (a-$K_xV_2O_5$) on one polarity of interdigital-patterned NP Au current microcollectors, followed by an annealing procedure that triggers the amorphous/crystalline dual-phase nanostructuring of the constituent a-$K_xV_2O_5$ via removing some crystal water sandwiched between $V_2O_5$ bilayers. Hereinto, the ac-$K_xV_2O_5$ not only creates new ion diffusion pathways and accommodation sites but keeps the large interface spacing of crystalline $V_2O_5$ bilayers (~10.8 Å) by making use of amorphous $K_{0.25}V_2O_5 \cdot nH_2O$ as molecular pillars. This unique nanoarchitecture is expected to substantially facilitate the intercalation of guest hydrated K[+] cations in the ac-$K_xV_2O_5$ (Fig. 1a), in sharp contrast with the K[+]-free ac-$V_2O_5$ that undergoes atrocious volume change and encounters sluggish diffusion kinetics due to the narrow interlayer spacing (Fig. 1b)[38,40,41,43]. Figure 1c illustrates a schematic of on-chip AR-PIMB full cells, which are monolithically constructed on glassy substrates by further integrating the c-$K_xMnO_2$ cathode on the other polarity of interdigital-patterned NP Au current microcollectors (NP Au/c-$K_xMnO_2$), with aqueous electrolyte of 0.5 M $K_2SO_4$. Therein, the current microcollectors built of 60 NP Au interdigitated microelectrodes (30 electrodes per polarity) are produced by chemically dealloying $Ag_{75}Au_{25}$ (at%) alloy precursor patterns with gaps of ~50 μm in $HNO_3$ solution (Fig. 1d)[46]. Each NP Au microelectrode has a 3D and bicontinuous nanoporous structure consisting of quasi-periodic Au ligaments and nanopore channels with a characteristic length of ~70 nm (Supplementary Fig. 1)[46]. Representative cross-sectional scanning electron microscope (SEM) images of AR-PIMB microelectrodes, i.e. the NP Au/ac-$K_xV_2O_5$ anode (Fig. 1e) and the NP Au/c-$K_xMnO_2$ cathode (Fig. 1f), demonstrate that both the electroactive materials are uniformly grown onto ~750-nm-thick nanoporous skeleton along Au ligaments.

The crystallographic structures of the constituent ac-$K_xV_2O_5$ are controlled by annealing treatment of the as-electrodeposited a-$K_xV_2O_5$ precursors, in which the hydrated K[+] cations with various components (x = 0, 0.1, 0.2, and 0.25) are introduced by tuning K[+] concentrations in the electrolytes (Supplementary

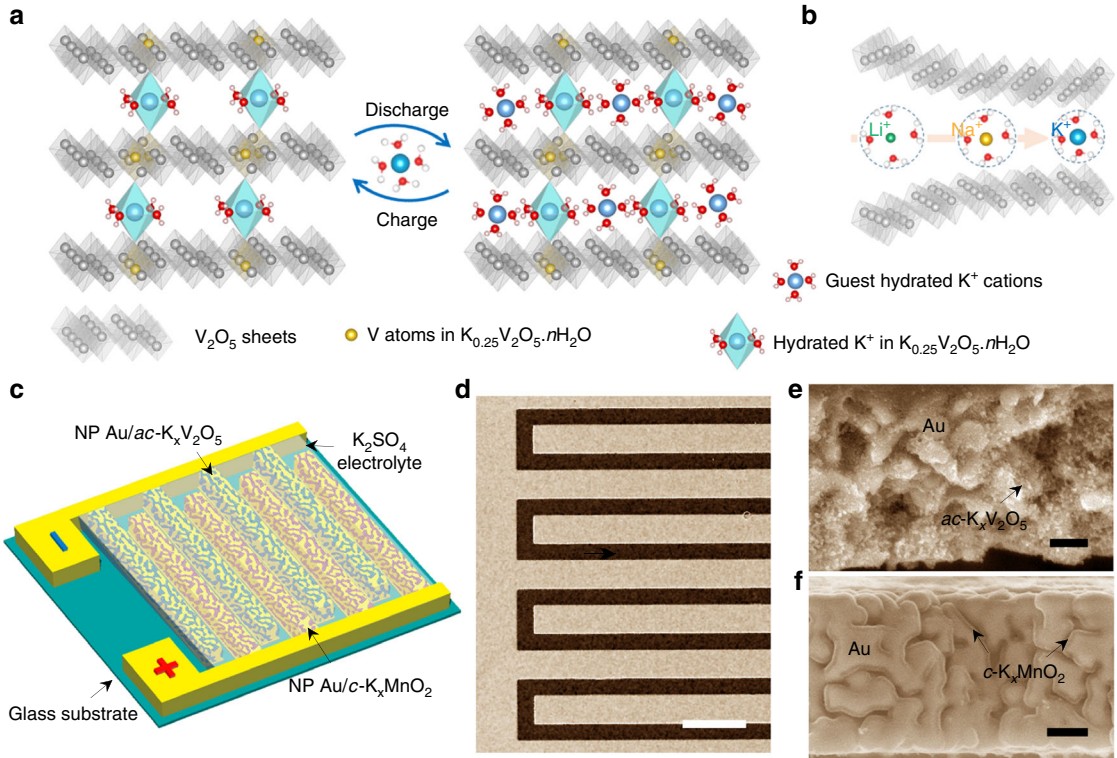

**Fig. 1** Schematic and microstructures of aqueous $K^+$ ion microbatteries. **a**, **b** Schematic diagrams illustrating the storage of hydrated ions in layered vanadium oxides: **a** $ac$-$K_xV_2O_5$ with a large interlayer spacing supported by $K_{0.25}V_2O_5 \cdot nH_2O$ molecular pillars anchored between $V_2O_5$ bilayers, and **b** orthorhombic $V_2O_5$ with a small interlayer spacing that substantially restricts the intercalation of hydrated $Li^+$, $Na^+$, and $K^+$ cations. **c** Schematic illustration of aqueous $K^+$-ion microbatteries, which are constructed by integrating $ac$-$K_xV_2O_5$ anode and $c$-$K_xMnO_2$ cathode on interdigitated nanoporous Au current collectors. **d** Low-magnification SEM image of interdigital-patterned nanoporous Au current collectors. **e**, **f** Cross-sectional SEM images for both anode and cathode of $K^+$-ion microbatteries. The anode (**e**) and cathode (**f**) are composed of layered $K_xV_2O_5$ and $K_xMnO_2$ supported by nanoporous Au current collectors, respectively. Scale bar, 200 μm (**d**), 200 nm (**e**, **f**)

Fig. 2, Supplementary Table 1). Along thermogravimetric analysis (TGA) and differential scanning calorimetry (DSC) behaviors (Supplementary Fig. 3), the NP Au/$a$-$K_xV_2O_5$ microelectrodes are annealed at 25, 200, and 300 °C for 12 h, respectively, wherein a crystallization phenomenon occurs with successively removing crystal water and structure water. As shown in X-ray diffraction (XRD) patterns for the representative $a$-$K_xV_2O_5$ ($x = 0.25$) annealed at 200 °C (Fig. 2a), there appear characteristic diffraction peaks corresponding to (101), (110), and (301) planes of the layer-structured $o$-$V_2O_5$ (JCPDS 41–1426)[40,44,47], apart from the bump one at $2\theta = 8.16°$ due to the (100) plane of the layered $a$-$K_xV_2O_5$ ($x = 0.25$)[37]. This implies the presence of amorphous/crystalline dual-phase nanostructure in the $ac$-$K_xV_2O_5$ with a interlayer spacing of ~10.8 Å, the value that is almost the same as that of the $a$-$K_xV_2O_5$ ($x = 0.25$) (Table 1). Typical high-resolution transmission electron microscope (HRTEM) image (Fig. 2c) with evident crystalline and amorphous fast Fourier transform (FFT) patterns (Fig. 2d, e) compellingly evidences the unique dual-phase nanostructure, in which the amorphous nanodomains are embedded in the crystalline $o$-$V_2O_5$ matrix. Meanwhile, the $ac$-$K_xV_2O_5$ displays the same characteristic Raman bands as the $a$-$K_xV_2O_5$ with the fingerprints of layer-type structure: the skeleton bent vibration at $163\ cm^{-1}$ and the stretching vibration of vanadyl V = O at $1015\ cm^{-1}$, attesting that it keeps the layered structure (Supplementary Fig. 4)[48,49]. When further increasing the annealing temperature to 300 °C, the $a$-$K_xV_2O_5$ is completely transformed into a crystalline mixture composed of the $o$-$V_2O_5$ and the monoclinic $K_{0.25}V_2O_5$ ($m$-$K_{0.25}V_2O_5$). As demonstrated by XRD patterns for the crystalline $K_xV_2O_5$ ($c$-$K_xV_2O_5$), the

additional diffraction peaks are ascribed to the (100), (002), (004), and (300) planes of the $m$-$K_{0.25}V_2O_5$ (JCPDS 39–0889) (Fig. 2a). Supplementary Figure 4c schematically illustrates the atomic structure of the $m$-$K_{0.25}V_2O_5$, where the $[V_4O_{12}]_n$ sheets consisting of V(1)$O_6$ and V(2)$O_6$ octahedra zigzag chains are linked by oxygen atoms to form 2D layered structure along the (001) plane, with $K^+$ cations intercalating between the layers. The $[V_4O_{12}]_n$ layers are further connected by V(3)$O_5$ and edge-sharing oxygen atoms to form a 3D tunnel structure[50]. Figure 2f shows the typical HRTEM image of the $c$-$K_xV_2O_5$, where the interplanar distances of 0.23 and 0.21 nm correspond to the lattice planes of the $m$-$K_{0.25}V_2O_5$(400) and the $o$-$V_2O_5$(002) (Fig. 2g), respectively. In view that the layered $o$-$V_2O_5$ is the primary phase in the $c$-$K_xV_2O_5$ (Fig. 2a), the interlayer spacing is evaluated to be ~4.42 Å according to the (001) diffraction peak position of the $o$-$V_2O_5$ (Table 1)[38,40,44]. In the structure evolution of the constituent $K_xV_2O_5$ ($x = 0.25$), X-ray photoelectron spectroscopy (XPS) analysis of oxygen-containing species justifies the change of water component sandwiched between the $V_2O_5$ bilayers. As shown in Fig. 2b for high-resolution O 1$s$ XPS spectra, there are three different oxygen species, i.e., the $O^{2-}$ in the $V_2O_5$ lattice, the $OH^-$, and the $H_2O$, to correspond to the peaks at the binding energies of 529.80, 530.90, and 533.01 eV[51]. Compared with that in the pristine $a$-$K_xV_2O_5$ (left plot in Fig. 2b), the O 1$s$ peak of $H_2O$ gradually attenuates in the $ac$-$K_xV_2O_5$ (middle plot of Fig. 2b), and then disappears in the $c$-$K_xV_2O_5$ (right plot of Fig. 2b) due to the loss of crystal water and structure water, respectively. Similar crystalline phenomenon also occurs in the $K^+$-free $a$-$V_2O_5$ to generate the $ac$-$V_2O_5$, but with a interlayer

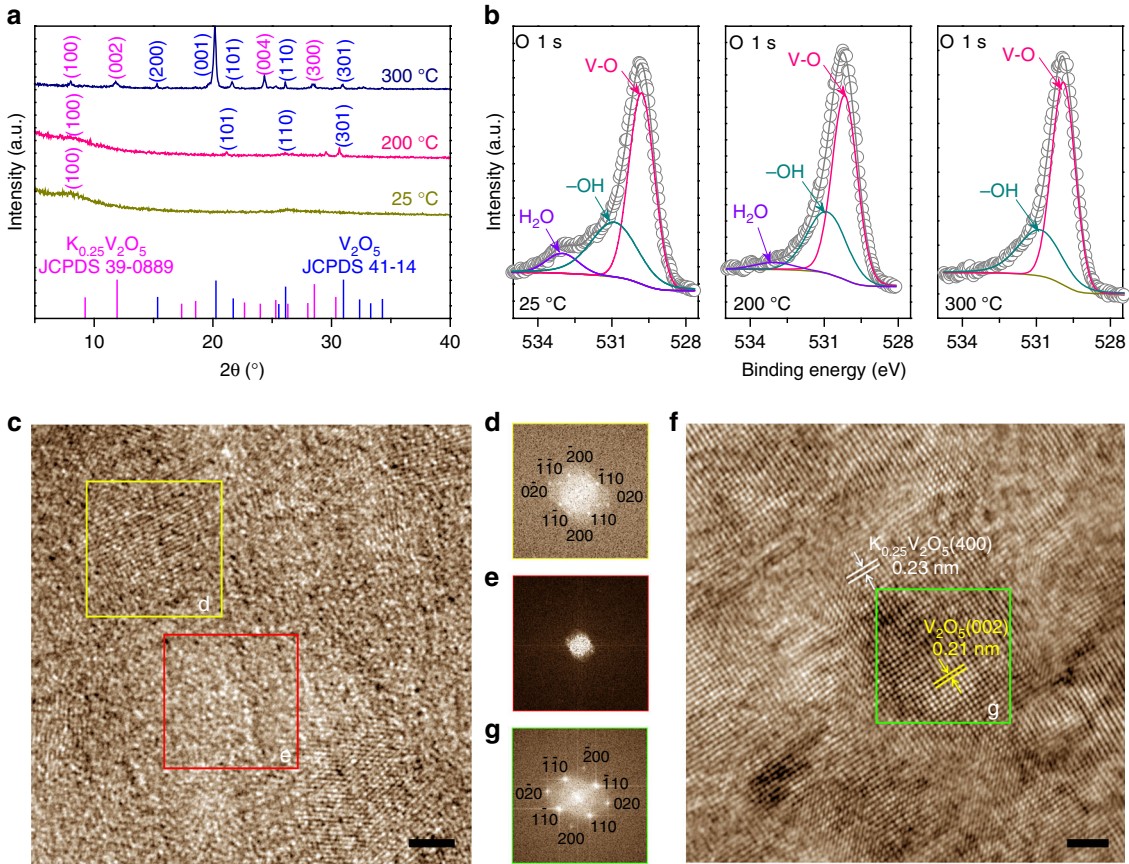

**Fig. 2** Microstructural characterization of the constituent $K_xV_2O_5$. **a** XRD patterns of layered $K_xV_2O_5$ ($x = 0.25$) that is annealed at 25, 200, and 300 °C. The line patterns show reference cards 39–0889 and 41–1426 for layered $K_{0.25}V_2O_5$ and $V_2O_5$ according to JCPDS. **b** High-resolution XPS spectra of O 1s in the layered $K_xV_2O_5$ ($x = 0.25$) that is annealed at 25, 200, and 300 °C. **c** HRTEM image of the layered $K_xV_2O_5$ ($x = 0.25$) with a dual-phase nanostructure that is produced by annealing at 200 °C. **d**, **e** FFT patterns of layered $K_xV_2O_5$ with crystalline region (yellow box) (**d**) and amorphous region (red box) (**e**). **f** HRTEM image of the layered $K_xV_2O_5$ ($x = 0.25$) with a complete crystallization at annealing temperature of 300 °C. **g** FFT pattern of layered crystalline $K_xV_2O_5$ (green box). Scale bar, 2 nm (**c**, **f**)

**Table 1 XRD and electrochemical results of layered $K_xV_2O_5$ and $V_2O_5$ supported by nanoporous Au current microcollectors**

| Materials | Interlayer space (Å) | Capacity (mAh cm$^{-3}$) | $R_I$ (Ω) | $R_{CT}$ (Ω) | $R_{IR}$ (Ω) | $Z_W$ (S s$^{1/2}$) |
|---|---|---|---|---|---|---|
| $a$-$K_{0.25}V_2O_5$ | 10.96 | 370 | 8.99 | 1.19 | 10.6 | 892.9 |
| $c$-$K_{0.25}V_2O_5$ | 4.42 | 500 | 10.79 | 2.19 | 16.1 | 741.9 |
| $ac$-$K_{0.25}V_2O_5$ | 10.83 | 717 | 5.53 | 0.28 | 6.6 | 3358.7 |
| $ac$-$K_{0.2}V_2O_5$ | 10.32 | 554 | 5.66 | 0.35 | 7.3 | 2198.2 |
| $ac$-$K_{0.1}V_2O_5$ | 5.33 | 464 | 5.77 | 0.61 | 7.8 | 1949.1 |
| $ac$-$V_2O_5$ | 4.36 | 242 | 19.97 | 0.94 | 21.8 | 1194.9 |

spacing directly decreasing to 4.36 Å as a consequence of squeezing out the crystal water at 200 °C (Table 1, Supplementary Figs. 5, 6)[38,40,44]. This observation is in sharp contrast with the $ac$-$K_xV_2O_5$, demonstrating the significant role of the amorphous $K_{0.25}V_2O_5 \cdot nH_2O$ as molecular pillars in maintaining large interlayer spacing. Owing to the decrease of the hydrated $K^+$ pillars, the interlayer spacing of $ac$-$K_xV_2O_5$ decreases with the $K^+$ concentration (Table 1, Supplementary Fig. 7).

**Electrochemical characterizations**. The electrochemical measurements of NP Au/$K_xV_2O_5$ microelectrodes are performed in 0.5 M $K_2SO_4$ aqueous electrolyte on the basis of a three-electrode configuration with a Pt foil as the counter electrode and an Ag/AgCl electrode as the reference electrode. Figure 3a shows typical cyclic voltammograms (CVs) of NP Au/$K_xV_2O_5$ ($x = 0.25$) microelectrodes within a voltage window of −0.8 to 0 V (versus

Ag/AgCl) at a scan rate of 50 mV s$^{-1}$. Relative to NP Au/$a$-$K_xV_2O_5$ and NP Au/$c$-$K_xV_2O_5$, the NP Au/$ac$-$K_xV_2O_5$ exhibits remarkably enhanced current density because of the unique dual-phase nanostructure of the constituent $ac$-$K_xV_2O_5$, which offers more room to accommodate guest hydrated $K^+$ cations and facilitate their solid-state diffusion kinetics in the pillar supported interlayer space. Associated with the nanoporous Au current microcollectors that enables the constituent $ac$-$K_xV_2O_5$ to be sandwiched between highly conductive ion and electron transport pathways, the CV curves at various scan rates from 5 to 1000 mV s$^{-1}$ retain a quasi-rectangular shape, indicating the exceptional high-rate capability of NP Au/$ac$-$K_xV_2O_5$ (Supplementary Fig. 8a, b). Although both NP Au/$a$-$K_xV_2O_5$ and NP Au/$c$-$K_xV_2O_5$ microelectrodes also benefit from such microelectrode architecture with fast electron and ion transports, they encounter rate-limited electrochemical energy-storage behaviors due to the poor accessibility and diffusion of

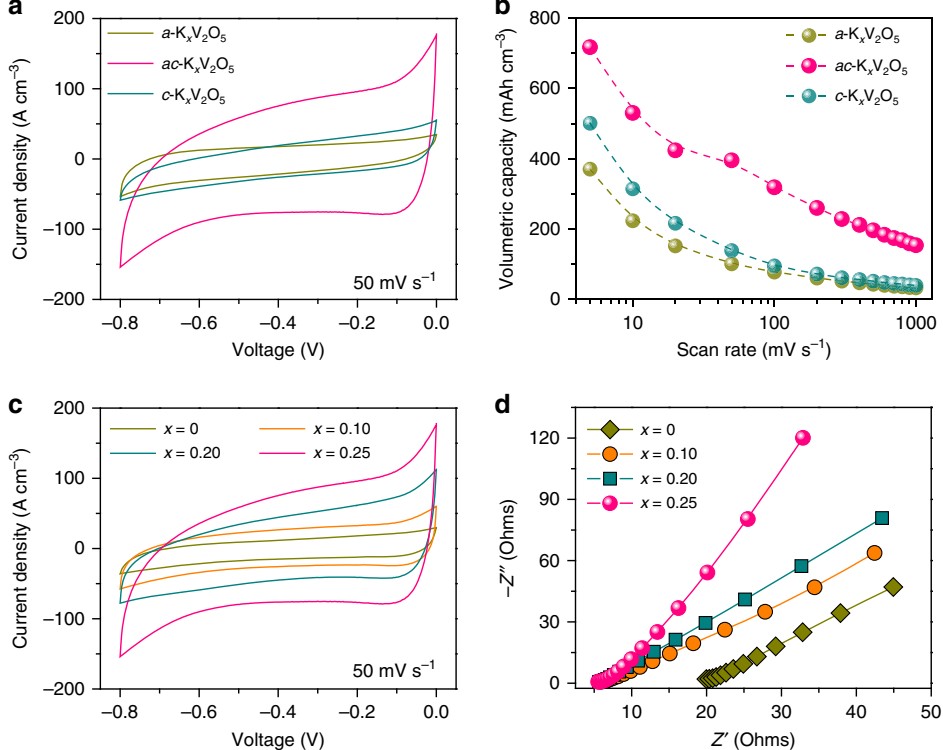

**Fig. 3** Electrochemical characterization of anodic nanoporous Au/$K_xV_2O_5$ microelectrodes. **a** Representative CV curves of nanoporous Au supported $a$-$K_xV_2O_5$, $ac$-$K_xV_2O_5$, and $c$-$K_xV_2O_5$ ($x = 0.25$) microelectrodes that are annealed at 25, 200, and 300 °C, respectively. Scan rate: 50 mV s$^{-1}$; electrolyte: 0.5 M $K_2SO_4$. **b** Volumetric capacities of nanoporous Au/$K_xV_2O_5$ ($x = 0.25$) microelectrodes with the constituent $a$-$K_xV_2O_5$, $ac$-$K_xV_2O_5$, and $c$-$K_xV_2O_5$ (annealed at 25, 200, and 300 °C) at various scan rates from 5 to 1000 mV s$^{-1}$. **c** Representative CV curves of nanoporous Au/$ac$-$K_xV_2O_5$ ($x = 0$, 0.1, 0.2, and 0.25) microelectrodes that are annealed at 200 °C. scan rate: 50 mV s$^{-1}$; electrolyte: 0.5 M $K_2SO_4$. **d** EIS spectra of nanoporous Au/$ac$-$K_xV_2O_5$ ($x = 0$, 0.1, 0.2, and 0.25) microelectrodes annealed at 200 °C

hydrated K$^+$ cations in the constituent electroactive materials, i.e., the $a$-$K_xV_2O_5$, in which too much crystal water fills in large interlayer spacing, and the $c$-$K_xV_2O_5$, of which the interlayer spacing is too narrow (Supplementary Fig. 8c, d). As a result, they exhibit much lower volumetric capacities than the NP Au/$ac$-$K_xV_2O_5$ at various scan rates (Fig. 3b). Therein, the gravimetric capacity of the constituent $ac$-$K_xV_2O_5$ reaches 382 mAh g$^{-1}$ at 5 mV s$^{-1}$ and retains 81 mAh g$^{-1}$ at 1000 mV s$^{-1}$, outperforming the constituent $a$-$K_xV_2O_5$ and $c$-$K_xV_2O_5$ (Supplementary Fig. 9a), as well as some of the best $V_2O_5$-based electrode materials in a full rate range reported previously for aqueous energy storage (Supplementary Fig. 9b). Electrochemical impedance spectroscopy (EIS) analysis in a frequency ranging from 100 kHz to 10 mHz demonstrates that the outstanding energy storage/delivery behavior of the constituent $ac$-$K_xV_2O_5$ results from the enhanced intercalation kinetics of hydrated K$^+$ cations in the dual-phase nanostructure. In the Nyquist plot, the EIS spectrum displays a quasi-semicircle with a ultrasmall diameter in the high- and middle-frequency ranges, followed by inclined lines in the low-frequency range (Supplementary Fig. 10a). At very high frequencies, the intercept at the real part represents the ohmic resistance ($R_I$) of the electrolyte and the intrinsic resistance of microelectrode; the semicircular behavior in the middle-frequency range corresponds to the charge transfer resistance ($R_{CT}$) and the double-layer capacitance ($C_F$); the slope of the inclined line at the low frequencies is the Warburg resistance ($Z_w$), which is a result of the frequency dependence of ion diffusion/transport inside the electrode. Using the complex nonlinear least-squares fitting method (Supplementary Fig. 10b), the NP Au/$ac$-$K_xV_2O_5$ is revealed to exhibit the lowest equivalent

series resistances ($R_I = 5.53\ \Omega$ and $R_{CT} = 0.28\ \Omega$) (Supplementary Fig. 10c) and the highest $Z_w$ value (3358 S s$^{1/2}$) (Supplementary Fig. 10d), reflecting the superior ion accessibility/diffusion and charge transfer in the dual-phase nanostructure during the electrochemical energy storage. The distinguished electrochemical behavior of the NP Au/$ac$-$K_xV_2O_5$ microelectrode is further elucidated by charge/discharge kinetics analysis with an assumption that the current density ($i$) obeys a power-law relationship with the scan rate ($v$)[52,53], namely $i = av^b$, where $a$ is an adjustable parameter, the $b$ value of 0.5 or 1 represents a diffusion- or surface-controlled process, respectively. In the $\ln(i)$−$\ln(v)$ plots (Supplementary Fig. 8e), the $b$ value of the NP Au/$ac$-$K_xV_2O_5$ is determined to be 0.86 due to enhanced insertion/extraction kinetics of the hydrated K$^+$ cations, in contrast with those of the NP Au/$a$-$K_xV_2O_5$ (0.60) and NP Au/$c$-$K_xV_2O_5$ (0.64) with evident limitations of poor accessibility and slow diffusion.

Figure 3c and Supplementary Fig. 11a–d compare the CV curves of NP Au/$ac$-$K_xV_2O_5$ microelectrodes, of which the constituent $ac$-$K_xV_2O_5$ has K$^+$ components changed from 0 to 0.25, at various scan rates. Owing to the maintenance of amorphous nanodomains of $K_{0.25}V_2O_5 \cdot nH_2O$ in the constituent $ac$-$K_xV_2O_5$, the NP Au/$ac$-$K_xV_2O_5$ microelectrodes exhibit evidently improved current densities compared with the one with the constituent K$^+$-free $ac$-$V_2O_5$, i.e., NP Au/$ac$-$V_2O_5$. Furthermore, the current density of NP Au/$ac$-$K_xV_2O_5$ microelectrode increases with the increasing $x$ value, which enlists the NP Au/$ac$-$K_xV_2O_5$ with $x = 0.25$ to achieve a volumetric capacity of as high as ~715 mAh cm$^{-3}$ at 5 mV s$^{-1}$ (Fig. 3b), about threefold the value of NP Au/$ac$-$V_2O_5$ (~242 mAh cm$^{-3}$) (Supplementary Fig. 11e, f). Even when increasing the scan rate

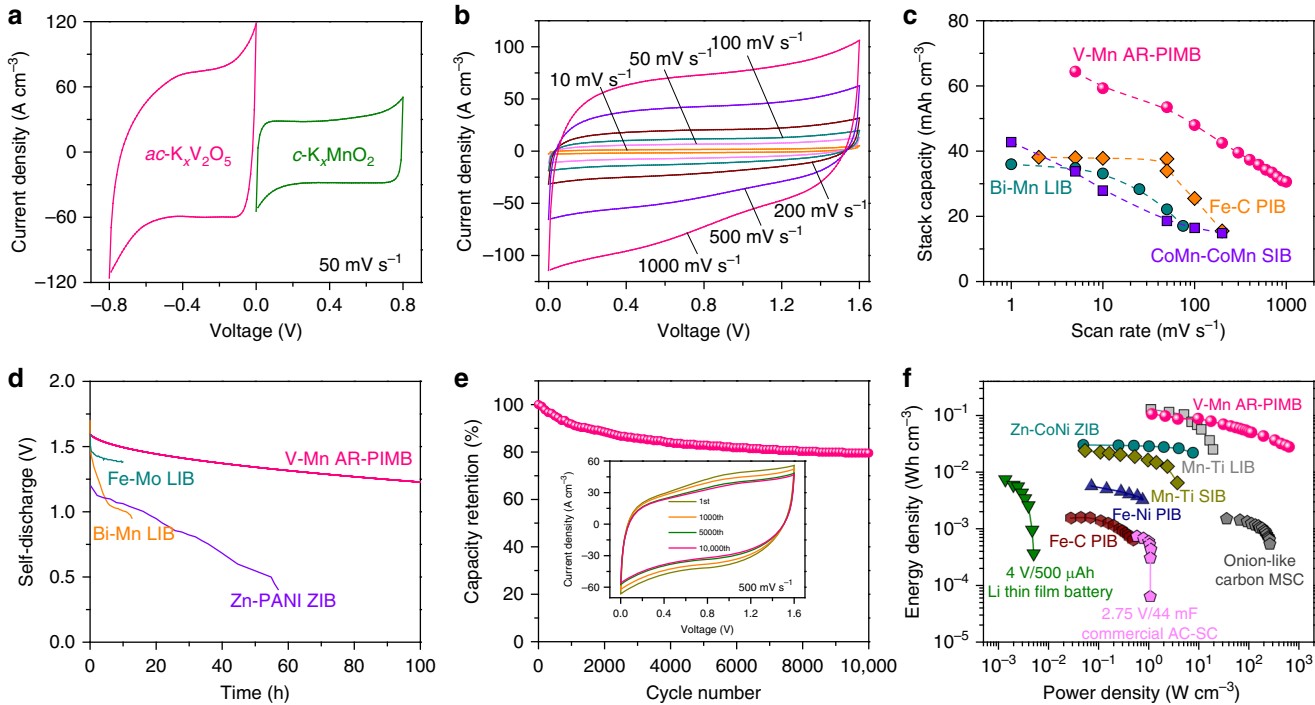

**Fig. 4** Electrochemical performance of V-Mn AR-PIMBs. **a** Representative CV curves for single electrode of $K_xV_2O_5$ ($x = 0.25$) anode and $K_xMnO_2$ ($x = 0.23$) cathode supported by nanoporous gold current collector in the voltage window of −0.8 to 0 and 0-0.8 V at a scan rate of 50 mV s$^{-1}$ in 0.5 M $K_2SO_4$. **b** CV curves at various scan rates for aqueous V-Mn AR-PIMBs that are constructed with $K_xV_2O_5$ and $K_xMnO_2$ on nanoporous gold current collectors as anode and cathode. Voltage window is extended to 1.6 V in aqueous electrolyte of 0.5 M $K_2SO_4$. **c** Stack capacity of aqueous V-Mn AR-PIMBs at various scan rates, comparing with previously reported rechargeable aqueous alkaline-metal-ion batteries, such as K$^+$-ion batteries with carbon-encapsulated $Fe_3O_4$ nanorod array anode and carbon nanotube film cathode in 3 M KOH electrolyte (Fe-C PIBs)[56], Li$^+$-ion batteries with $Bi_2O_3$ anode, and $LiMn_2O_4$ cathode in a mixed electrolyte of $Li_2SO_4$ and LiCl (Bi–Mn LIBs)[57], and symmetric Na$^+$-ion batteries of biphase cobalt–manganese oxide nanosheets in 0.1 M $Na_2SO_4$ (CoMn–CoMn SIBs)[21]. **d** Comparison of self-discharge performance for aqueous V-Mn AR-PIMBs with aqueous Bi–Mn LIBs[57], $LiFePO_4$//$Mo_6S_8$ (Fe–Mo) LIB, and Zn//polyaniline Zn$^{2+}$-ion battery (Zn–PANI ZIB)[58]. **e** Cycling stability of aqueous V-Mn K$^+$-ion microbatteries at the scan rate of 500 mV s$^{-1}$. Inset: Typical CV curves at different cycles. **f** Ragone plot comparing stack energy and power densities of V-Mn AR-PIMB with commercially available 4 V/500 μAh Li thin-film battery, 2.75 V/44 mF commercial activated carbon supercapacitors (AC-SC)[9,14], Fe–C PIB[56], carbon fiber supported $Fe_3O_4$//NiO (Fe–Ni) PIB, $Na_{0.44}MnO_2$//$NaTi_2(PO_4)_3$ (Mn–Ti) SIB[28], $Li_{1.1}Mn_2O_4$//$LiTi_2(PO_4)_3$ (Mn–Ti) LIB and $CoNi(OH)_2$//Zn (CoNi-Zn) ZIB[59], as well as onion-like carbon-based microsupercapacitors (onion-like carbon MSC)[9]

to 1000 mV s$^{-1}$, it still retains the capacity of 153 mAh cm$^{-3}$, which is more than six times higher than the NP Au/$ac$-$V_2O_5$ (Supplementary Fig. 11e). These facts demonstrate the important roles of amorphous $K_{0.25}V_2O_5·nH_2O$ nanodomains in improving energy-storage behaviors of the $ac$-$K_xV_2O_5$, i.e., serving as molecular pillars to support large interlayer spacing for facilitating the accessibility and diffusion of guest hydrated K$^+$ cations[29,30], and acting as dopants to ameliorate the electronic conductivity of $V_2O_5$ bilayers for boosting electron transfer[54,55]. This is further demonstrated by their distinct EIS spectra in the Nyquist plots (Fig. 3d). As shown in Supplementary Fig. 12a, all the NP Au/$ac$-$K_xV_2O_5$ ($x$ = 0.10, 0.20, and 0.25) microelectrodes have much lower intrinsic electrode resistances ($R_I$ = ~5.5–5.7 Ω) and charge transfer resistances ($R_{CT}$ = ~0.28–0.61 Ω) than the NP Au/$ac$-$V_2O_5$ ($R_I$ = ~19.9 Ω and $R_{CT}$ = ~0.94 Ω), despite slight variations as a function of K$^+$ component. While for the Warburg slope reflecting the ion diffusion kinetics (Supplementary Fig. 12b), it increases with the component of the hydrated K$^+$ pillars and reaches the highest value for the NP Au/$ac$-$K_xV_2O_5$ ($x$ = 0.25).

**Electrochemical performance of microdevices**. In view of the outstanding electrochemical properties of NP Au/$ac$-$K_xV_2O_5$ ($x$ = 0.25) microelectrodes, they are used as anodes to pair with the NP Au/$c$-$K_xMnO_2$ cathodes (Supplementary Fig. 13) to build

full microcells of on-chip V-Mn AR-PIMBs when integrating the $c$-$K_xMnO_2$ on the other polarity of interdigitated NP Au current microcollectors (Fig. 1c–f). By virtue of the potential difference between the NP Au/$ac$-$K_xV_2O_5$ anode (−0.8 to 0 V) and the NP Au/$c$-$K_xMnO_2$ cathode (0–0.8 V) (Fig. 4a), the working voltage window of full microcells is extended to 1.6 V in the aqueous electrolyte of 0.5 M $K_2SO_4$. Figure 4b shows CV curves of the V-Mn AR-PIMBs at various scan rates corresponding to a full charge or discharge time from ~1.6 s to ~5.3 min, during which they are able to achieve a stack capacity of as high as ~64 mAh cm$^{-3}$ based on the total volume of the full microcells. About half of the stack capacity (31 mAh cm$^{-3}$) can be retained at exceptionally high discharge rates of 1000 mV s$^{-1}$ (namely, full discharge in less than 2 s) (Fig. 4c). This exceptional rate performance enlists the V-Mn AR-PIMBs to outperform some of the best rechargeable aqueous metal-ion batteries reported previously: such as aqueous K$^+$-ion batteries with carbon-encapsulated $Fe_3O_4$ nanorod array anode and carbon nanotube film cathode in 3 M KOH electrolyte (Fe–C PIBs)[56], Li$^+$-ion batteries with $Bi_2O_3$ anode, and $LiMn_2O_4$ cathode in a mixed electrolyte of $Li_2SO_4$ and LiCl (Bi–Mn LIBs)[57], as well as symmetric Na$^+$-ion batteries of biphase cobalt–manganese oxide nanosheets in 0.1 M $Na_2SO_4$ (CoMn–CoMn SIBs)[21]. Meanwhile, the V-Mn AR-PIMBs displays very low self-discharge, with less than 3.6 mV h$^{-1}$, much better than LIBs assembled with $Mo_6S_8$

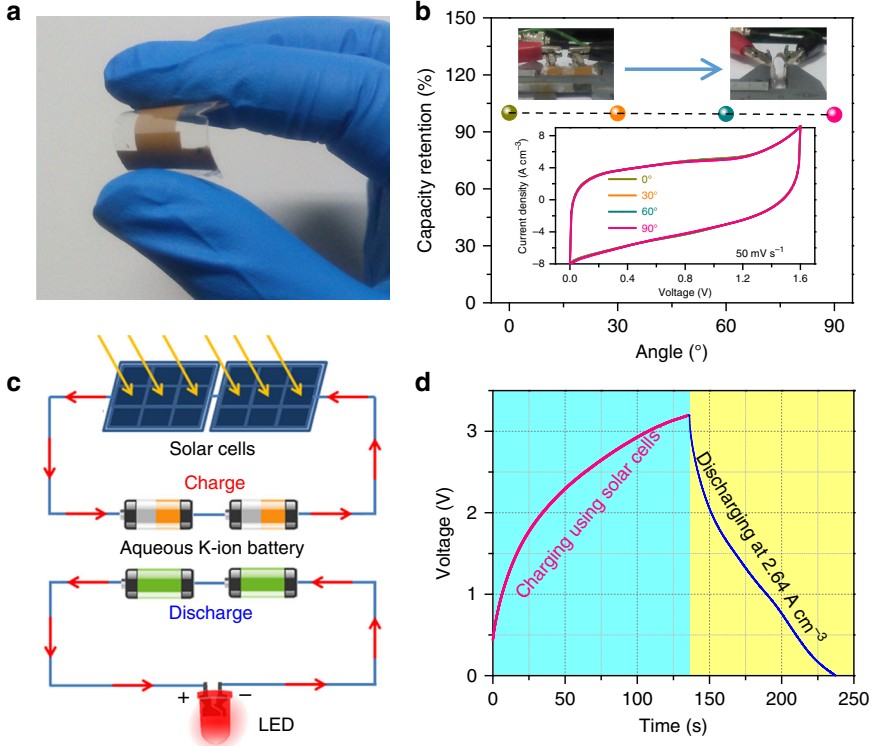

**Fig. 5** Flexible all-solid-state V-Mn AR-PIMBs integrated in solar cells. **a** Typical photograph for flexible all-solid-state V-Mn AR-PIMBs supported by PMMA substrates with PVA/KCl gel electrolyte. **b** Capacity retention of flexible aqueous V-Mn AR-PIMBs at different macroscopic bend angles. Inset: Photographs of flexible all-solid-state V-Mn K$^+$-ion microbatteries bended from 0 to 90° (top) and their corresponding CV curves (bottom). **c** Schematic showing a self-powered system constructed with two-series flexible V-Mn AR-PIMBs, 2 V thin-film solar cells and LED lights: when sunlight is available, the V-Mn AR-PIMBs are charged by solar cells (top); when sunlight is unavailable, they are discharged to power LED lights (bottom). **d** Typical charge/discharge curves for the charged process powered by solar cells and the discharge process at a current density of 2.64 A cm$^{-3}$

anode and LiFePO$_4$ cathode (Mo-Fe LIBs) in Li-TFSI: H$_2$O = 1 (21.1 mV h$^{-1}$), Bi–Mn LIBs (58.9 mV h$^{-1}$)[58], and Zn$^{2+}$-ion batteries with Zn anode and polyaniline film cathode (Zn–PANI ZIBs) in 1 M ZnCl$_2$ (12.7 mV h$^{-1}$) (Fig. 4d). Figure 4e shows that the cycling performance of the V-Mn AR-PIMB, which retains 80% of its initial capacity after 10,000 CV cycles at a scan rate of 500 mV s$^{-1}$ (inset of Fig. 4e), demonstrates the impressive long-term stability in the voltage window between 0 and 1.6 V.

In the Ragone plot, Fig. 4f compares the stack-volumetric power and energy densities of V-Mn AR-PIMB with the values of commercially available devices including 4 V/500 μAh Li thin-film battery and 2.75 V/44 mF activated carbon supercapacitors[9,14]. Our microbattery has a volumetric energy density of ~103 mWh cm$^{-3}$ based on the whole devices volume, which is 14-fold higher than 4 V/500 μAh Li thin-film battery, when delivering at the electrical power (1.2 W cm$^{-3}$) compared to 2.75 V/44 mF activated carbon supercapacitors. As a result of monolithic integration of on-chip V-Mn AR-PIMB constructed on 3D bicontinuous nanoporous Au current microcollectors, which minimizes the internal resistance and facilitates rapid ion transport, the electrical power of V-Mn AR-PIMB can reach as high as ~600 W cm$^{-3}$ with the energy density of ~27 mWh cm$^{-3}$, much higher than micro-supercapacitors based on onion-like carbon[9]. To our knowledge, the volumetric energy and power densities of V-Mn AR-PIMBs is the highest value among recently reported aqueous alkaline-metal-ion batteries on the basis of Li$^+$[20], Na$^+$[26], K$^+$[56], and Zn$^{2+}$[59] to date (Fig. 4f).

To meet specific micropower needs for portable or wearable electronics, all-solid-state flexible V-Mn AR-PIMBs are further developed by using a gel electrolyte of polyvinyl alcohol (PVA)/KCl and shifting on polymethyl methacrylate (PMMA)

substrates (Fig. 5a). Because of the outstanding mechanical flexibility (inset of Fig. 5b), the all-solid-state microbatteries achieve almost the same capacity when bended from 0° to 90° (Fig. 5b). Furthermore, they can be connected in series to increase the working voltage and in parallel to store more energy, showing potential applications as stand-alone micropower sources or intermediate energy-storage components in electronic and optoelectronic microdevices. As demonstrated by a self-powered system that is constructed with 2-V thin-film solar cells to harvest energy from renewable and sustainable sunlight (Supplementary Fig. 14a), two V-Mn AR-PIMBs in series are charged to store energy and then discharged to power LED light when sunlight is available and unavailable (Fig. 5c). Figure 5d displays their typical voltage–time profile during the processes of charge and discharge in a working voltage window of 0–3.2 V. When solar cells are irradiated by sunlight, the voltage of the two-series V-Mn AR-PIMBs increases to 3.2 V in 136 s, meaning that the microbatteries can be charged successfully. While sunlight is unavailable, the V-Mn AR-PIMBs are discharged for 100 s at 2.64 A cm$^{-3}$, powering the LED light (Supplementary Fig. 14b).

## Discussion

In summary, we have demonstrated representative layered TMOs, K$_x$V$_2$O$_5$, with an amorphous/crystalline dual-phase nanostructure, as promising anode materials of AR-PIMB when they are integrated on 3D bicontinuous nanoporous metal current microcollectors. Dual-phase nanostructuring of the layered $ac$-K$_x$V$_2$O$_5$ is to remove some crystal water and use the amorphous K$_{0.25}$V$_2$O$_5$·$n$H$_2$O as molecular pillars to support the large inter-layer distance, which creates additional room in interlayer space

to accommodate guest hydrated $K^+$ cations with significantly enhanced accessibility/diffusion kinetics. Associated with the unique architecture of 3D bicontinuous nanoporous metal-based current microcollectors, which simultaneously facilitate electron and ion transports in the whole microelectrodes, the NP Au/$ac$-$K_xV_2O_5$ exhibits exceptionally energy storage with volumetric capacity of as high as ~715 mAh cm$^{-3}$, and high rate capability. The outstanding electrochemical properties enlist AR-PIMB, which are constructed by integrating the $ac$-$K_xV_2O_5$ anode and the $c$-$K_xMnO_2$ cathode on interdigitated 3D nanoporous Au current microcollectors, to really realize energy storage with lithium-ion microbattery-like capacity and supercapacitor-like rate performance and long-term cycling stability. Their maximum energy and power densities reach ~103 mWh cm$^{-3}$ and ~ 600 W cm$^{-3}$, outperforming not only commercially available energy-storage devices, such as 4 V/500 μAh Li thin-film battery and 2.75 V/44 mF activated carbon supercapacitors, but also previously reported aqueous alkaline ion batteries based on Li$^+$, Na$^+$, and K$^+$, in addition to Zn$^{2+}$. The superior performance makes them promising candidates as next-generation micro-power sources for satisfying the demand of high-energy and -power densities in miniaturized portable electronics and MEMS.

## Methods

**Fabrication of aqueous potassium ion microbatteries**. Microbattery devices were constructed by electrodepositing vanadium oxides and manganese oxides as anode and cathode, respectively, on interdigital-patterned Au current microcollectors with a 3D bicontinuous nanoporous architecture. Pure Cr (~50-nm-thick), Au (~100-nm-thick) and Ag$_{75}$Au$_{25}$ (at %) alloy (~750-nm-thick) films were consecutively sputtered on micropatterned molds consisting of 30 pairs of interdigital channels with the width of 50 μm and the gap of 100 μm, which were firstly fabricated on glass substrates by a general lithography technology. After stripping the micropatterned mold in 5 wt% NaOH, the metallic patterns were annealed in Ar air at 500 °C for 2 h and then chemically dealloyed in concentrated HNO$_3$ for 15 min. Electrodeposition of electroactive TMOs on the as-prepared interdigital nanoporous Au patterns were carried out on a classic three-electrode setup (Iviumstat electrochemical analyser; Ivium Technology), in which Pt foil and Ag/AgCl electrode were employed as the counter electrode and the reference electrode, respectively. For the anodes, potassium vanadium oxide ($K_xV_2O_5 \cdot nH_2O$) was electrodeposited by a potentiodynamic method in a pH = 1.8 electrolyte containing 100 mM VOSO$_4$, 30 mL pure water and 70 mL ethanol, as well as K$_2$SO$_4$ with various concentrations from 0 to 30 mM, followed by annealing at 25, 200, and 300 °C under ambient conditions for 12 h. Based on the total transfer charge in the electrodepositing process, the loading density of $K_xV_2O_5 \cdot nH_2O$ is calculated to be 1.88 g cm$^{-3}$. While for the cathode, the cryptomelane $K_xMnO_2 \cdot nH_2O$ was incorporated into the nanoporous Au microelectrodes by a pulse electrodeposition technology in 0.1 M KMnO$_4$ neutral electrolyte. The loading mass of these TMOs was controlled by the transferred charges during the electrodeposition processes based on an assumption of 100% efficiency.

To develop all-solid-state flexible potassium ion microbatteries, PMMA film was deposited on the as-prepared nanoporous Au current microcollectors by dispersing 2 mL PMMA solution (4 wt% in ethyl lactate) and then baking on a hotplate at 80 °C for 30 min. After etching the Cr layer in 0.5 M (NH$_4$)$_2$Ce(NO$_3$)$_6$ solution, the PMMA supported nanoporous gold micropatterns was exfoliated and incorporated with the $ac$-$K_xV_2O_5$ anode and the $c$-$K_xMnO_2$ cathode by the same procedures as that in the fabrication of rechargeable aqueous potassium ion microbatteries. Using PVA/KCl gel electrolyte, all-solid-state flexible potassium ion microbatteries are fabricated and sealed by PDMS for electrochemical test.

**Microstructural and chemical characterizations**. The microstructure and chemical composition of the anodic and cathodic electrodes in the microbatteries were characterized using a field-emission scanning electron microscope (JEOL JSM-6700F, 15 keV) equipped with an X-ray energy-dispersive spectroscopy and a field-emission transmission electron microscope (JEOL JEM-2100F, 200 keV). XRD measurements were carried out on a D/max2500pc diffractometer using Cu $K_\alpha$ radiation. XPS was performed on an ESCAlab220i-XL electron spectrometer from VG Scientific using Al $K_\alpha$ radiation. TGA and DSC were measured on a DSC 7 analyzer at the ramping rate of 10 °C min$^{-1}$ in Ar air. Raman spectra were collected on a micro-Raman spectrometer (Renishaw) with a laser wavelength of 532 nm at 1 mW.

**Electrochemical measurements**. Electrochemical properties of single anodic and cathodic electrodes were characterized in 0.5 M K$_2$SO$_4$ aqueous electrolyte on a classic three-electrode setup with Pt foil as the counter electrode and Ag/AgCl as

the reference electrode. Electrochemical behaviors of battery microdevices were measured by a two-electrode setup in 0.5 M K$_2$SO$_4$ aqueous electrolyte. Cycling stability test of potassium ion microbatteries was performed in a voltage window from 0 to 1.6 V at a scan rate of 500 mV s$^{-1}$ for over 10000 cycles. Self-discharge measurements were carried out by charging aqueous potassium ion microbattery to 1.6 V at 110 μA, followed by open-circuit potential self-discharging for 100 h.

## Data availability

All relevant data are available from the corresponding authors upon request.

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

## Acknowledgements

This work was supported by National Natural Science Foundation of China (Nos. 51871107, 51631004), Top-notch Young Talent Program of China (W02070051), Chang Jiang Scholar Program of China (Q2016064), the Program for JLU Science and Technology Innovative Research Team (JLUSTIRT, 2017TD-09), the Fundamental Research Funds for the Central Universities, and the Program for Innovative Research Team (in Science and Technology) in University of Jilin Province.

## Author contributions

X.-Y.L., Q.J. and Y.-Q.L. conceived and designed the experiments. Y.-Q.L., H.S., S.-B.W., Y.-T.Z., Z.W. and X.-Y.L. carried out the fabrication and analysis of materials and performed the electrochemical and microstructural characterizations. X.-Y.L., Q.J. and Y.-Q.L. wrote the paper, and all authors discussed the results and commented on the manuscript.

## Additional information

**Competing interests:** The authors declare no competing interests.

