## [Peer Review File · Nature Communications]

Reviewers' comments:

Reviewer #1 (Remarks to the Author):

This work presents a fabrication of an asymmetric supercapacitor device using amorphous/crystalline dual-phase $K_xV_2O_5$ anode materials and MnO_2 based cathode materials in an aqueous K-ion electrolyte. The authors suggested that such dual-phase $K_xV_2O_5$ with pre-intercalated hydrated K^+ cations acting as molecular pillars would provide a relative large interlayer spacing for accommodating K^+ cations and facilitating their diffusion kinetics. Thus, the resulting supercapacitor devices demonstrated high volumetric performance. While this is a clearly written communication presenting a great amount of information, it is not of sufficient novelty and impact to warrant publication in Nature Communications.

Some specific comments are as followings:

- (i) It is not obvious that the authors had provided the loading density of the electrode, which makes the reviewer or readers difficult to compare the reported performance with the published results (in gravimetric performance). Since V_2O_5 is one of the most studied electrode materials for aqueous storage, it is unclear whether the reported dual-phase electrodes are advantageous or not.
- (ii) It seems the pre-intercalated water play an important role in determining the optimum structure of the electrode, TGA measurement showing water content will be necessary (only DSC was shown in Figure s3),
- (iii) Authors believed that pre-intercalated hydrated K^+ cations acting as molecular pillars would provide a relative large interlayer spacing for K-ion storage, however, XRD patterns (Figure 2a and s4) didn't suggest the formation of a layered structure.
- (iv) The authors showed a large potential window of 3.2 V (Figure 5d) of the asymmetric cell using a K_2SO_4 electrolyte, it is far beyond the water decomposition potential window of 1.23 V. Will electrolyte (particularly water) decompose to H_2 and O_2 in such conditions?

Reviewer #2 (Remarks to the Author):

The referee thinks that the paper should be resubmitted to more specialized journals dealing with battery technology and electrochemistry because of insufficient data and unclear discussion on experimental results.

Reviewer #3 (Remarks to the Author):

Li et al. reported a brand-new research strategy for aqueous rechargeable potassium-ion microbatteries (AR-PIMBs). The work innovatively explored the dual-phase $K_xV_2O_5$ nanostructure on Au current microcollectors, which achieved long-term cycling stability. The XRD and TEM prove the dual-phase of $K_xV_2O_5$ nanostructure. On the other hand, the potassium-ion microbatteries constructed with $K_xV_2O_5$ anode and K_xMnO_2 cathode present an excellent performance. The flexible all-solid-state V-Mn AR-PIMBs integrated in solar cells also exhibited an outstanding performance. It is a clear and solid work. I think the paper will lead a new research direction and should be of interest to the readers. I recommend accepting it in Nature Communication after minor revision.

1. The authors believed that Au current microcollectors are important to form dual-phase $K_xV_2O_5$ nanostructure. Thus, how about other current collectors? For example Cu foil?
2. In Figure 2, the letter should be rearranged.
3. In Figure 3d and Figure S7, is the EIS only straight line? The authors should give more

explanation.

4. How to increase the value of x of $K_xV_2O_5 \cdot nH_2O$? Is it able to achieve $x=3$ or higher value?
5. Some sentences are too long, and needed to be simplified. For example, in Abstract, line 14-19; Line 86-92
6. Some new and important publications in K/Na ion batteries should be referred such as Angew. Chem. Int. Ed. 2018, 57, 8540; Advanced Materials, 2017, 29 (24), 1700989; Advanced Materials, 2017, 29(1), 1604007; Angew. Chem. Int. Ed., 2019, 58, 1484-1488.

Response to reviewers' comments

Reviewer #1 (Remarks to the Authors)

This work presents a fabrication of an asymmetric supercapacitor device using amorphous/crystalline dual-phase $K_xV_2O_5$ anode materials and MnO_2 based cathode materials in an aqueous K-ion electrolyte. The authors suggested that such dual-phase $K_xV_2O_5$ with pre-intercalated hydrated K^+ cations acting as molecular pillars would provide a relative large interlayer spacing for accommodating K^+ cations and facilitating their diffusion kinetics. Thus, the resulting supercapacitor devices demonstrated high volumetric performance.

Reply: We thank the reviewer for finding potential interest of our work. We also appreciate the reviewer for the constructive comments, according to which we have carried out supplementary experiments, including DTA measurement and Raman spectroscopy characterizations of the *ac*- $K_xV_2O_5$ supported by NP Au current collectors. In addition, we also have re-evaluated the gravimetric capacities of the constituent *ac*- $K_xV_2O_5$ to compare with the values of the best V_2O_5 -based electrodes reported previously. Based on these results, we have comprehensively revised the manuscript. The details can be found below.

While this is a clearly written communication presenting a great amount of information, it is not of sufficient novelty and impact to warrant publication in Nature Communications.

Reply: We firstly appreciate the reviewer for approving our clear presentation with "a great amount of information" on the amorphous/crystalline dual-phase nanostructure of $K_xV_2O_5$, which essentially boosts intercalation/diffusion kinetics of hydrated K^+ cations for high-density energy storage/delivery at fast charge/discharge rates.

Aqueous rechargeable microbatteries are novel energy-storage microdevices that hold great promise to serve as stand-alone micropower sources or miniaturized energy-storage components complementing energy conversion devices (such as solar cells) for safe and reliable energy autonomy in self-powered portable microelectronics and wireless sensor networks. By virtue of battery chemistries based on electrochemical intercalation/de-intercalation of hydrated alkaline cations in aqueous electrolytes, the aqueous rechargeable microbatteries are expected to achieve lithium-ion microbattery-like capacity and microsupercapacitor-like rate performance and circumvent the safety

issues associated with highly toxic and flammable organic electrolytes, distinguished from traditional lithium-ion microbatteries and microsupercapacitors. In view of the K^+ cations having the smallest hydrated radii and the highest conductivity in water-based electrolytes, in this work, we construct monolithic aqueous rechargeable potassium-ion microbatteries (AR-PIMBs) on the basis of interdigital-patterned metal current microcollectors with a three-dimensional nanoporous architecture, in which anodic and cathodic layered metal oxides are integrated as electroactive materials. Although layered V_2O_5 as electroactive materials have been extensively investigated because of low cost, high Earth abundance and high theoretical capacity (443 mAh g^{-1}), the superior practical capacity and rate capability have not been fully exploited for practical applications in aqueous electrolytes. In comparison with electrodes based on V_2O_5 and its composites with various low-dimensional nanostructures (nanoparticles, nanowires and nanobelts), we would like to address that our monolithic NP Au/*ac*- $K_xV_2O_5$ microelectrodes are intrinsically different from all of them in both structure and physical/chemical properties. In the previous reports on V_2O_5 -based electrodes, they are usually constructed by immobilizing low-dimensional nanostructures of V_2O_5 or its composites on planar current collectors via polymer binder. Such electrode structure always leads to undesirably low conductance although there are some conductive carbon materials, such as activated carbon, carbon nanotubes and graphene, to be introduced to facilitate electron transfer. This is due to short electron transport in the nanomaterials as well as high contact resistances within nanocomposite blocks capped with insulating surfactant molecules and V_2O_5 layer, and between current collectors and electrodes. In contrast, our monolithic NP Au/*ac*- $K_xV_2O_5$ microelectrodes have the constituent *ac*- $K_xV_2O_5$ to be uniformly and directly incorporated into three-dimensional and bicontinuous nanoporous metal current microcollectors via the formation of abundant and stable metal/oxide interfaces. The unique microelectrode architecture essentially ameliorates transport kinetics of electrons and ions by offering them pathways along metal ligaments and nanopore channels, respectively. At the same, the abundant metal/oxide interfaces not only facilitate the electron transfer from the electroactive *ac*- $K_xV_2O_5$ to the current collectors but steadily confine the *ac*- $K_xV_2O_5$ in the nanopore channels of NP Au for significantly enhanced mechanical/electrochemical durability, as demonstrated by **Figure 4e** and **Figure 5a,b**. Furthermore, we develop a facile annealing strategy to trigger the amorphous/crystalline dual-phase nanostructuring of the constituent *ac*- $K_xV_2O_5$. The dual-phase nanostructured *ac*- $K_xV_2O_5$ keeps large interlayer spacing via pre-intercalated hydrated-potassium cations

as molecular pillars and removes secondary-bound interlayer water to create new channels, for accommodating more guest hydrated K^+ cations and facilitating their diffusion kinetics. This is essentially distinguished from previously reported layered V_2O_5 , in which too many crystalline water molecules sandwiched in interlayer spacing block the intercalation of hydrated cations. Even though some strategies have been proposed to reduce or remove the crystalline water for weakening the blocking influence on the intercalation of hydrate cations, crystallization usually takes place to produce the orthorhombic V_2O_5 (ρ - V_2O_5) with small interlayer spacing. This not only fails to improve the specific capacity and rate capability but leads to extra massive volume change during the insertion/extraction of hydrated cations to limit microbattery lifetime. Compared with some of the best V_2O_5 -based electrodes previously reported, the constituent ac - $K_xV_2O_5$ exhibits much higher specific capacity (as high as 382 mAh g^{-1} , ~86% of the theoretical value) with exceptional rate capability, as shown in supplementary **Figure S9**. Moreover, we further develop high energy- and power-density AR-PIMBs and their flexible microdevices as micropower sources to complement other energy conversion devices such as commercial solar cells.

Regarding the novelty and impact of our work, as pointed out by Reviewers, "*The work innovatively explored the dual-phase $K_xV_2O_5$ nanostructure on Au current microcollectors*" (the comment of Reviewer #3), "*such dual-phase $K_xV_2O_5$ with pre-intercalated hydrated K^+ cations acting as molecular pillars would provide a relative large interlayer spacing for accommodating K^+ cations and facilitating their diffusion kinetics. Thus, the resulting supercapacitor devices demonstrated **high volumetric performance**.*" (this Reviewer). "*Li et al. reported a **brand-new** research strategy for aqueous rechargeable potassium-ion microbatteries (AR-PIMBs)*" "I think **the paper will lead a new research direction** and should be of interest to the readers" (Reviewer #3). Actually, both the approach to prepare NP Au/ ac - $K_xV_2O_5$ microelectrodes with the unique dual-phase nanostructure and the aqueous rechargeable potassium-ion microbatteries have never been reported before. More importantly, the novel ac - $K_xV_2O_5$ supported by interdigital-patterned NP Au current microcollectors shows superior gravimetric/volumetric capacity, high rate capability and long-term stability ("*the resulting supercapacitor devices demonstrated high volumetric performance*", the comment from this reviewer; "*which achieved long-term cycling stability*", the comment from Reviewer #3) relative to some of the best V_2O_5 -based electrodes previously reported (supplementary **Figure S9**). These predominant properties enable the AR-PIMBs to exhibit high energy and power densities (**Figure 4f**) ("*On the other hand, the*

potassium-ion microbatteries constructed with $K_xV_2O_5$ anode and K_xMnO_2 cathode present an excellent performance. The flexible all-solid-state V-Mn AR-PIMBs integrated in solar cells also exhibited an outstanding performance.", the comment from Reviewer #3).

In view of the novelty, excellent performances and scientific implications of the nanoporous metal/oxide hybrid microelectrodes and the aqueous rechargeable potassium-ion microbatteries, we wish the reviewer could share our confidence and belief that the work reported in this paper deserves to be published in a high-impact journal, like *Nature Communications*.

(1) It is not obvious that the authors had provided the loading density of the electrode, which makes the reviewer or readers difficult to compare the reported performance with the published results (in gravimetric performance). Since V_2O_5 is one of the most studied electrode materials for aqueous storage, it is unclear whether the reported dual-phase electrodes are advantageous or not.

Reply: We thank the reviewer for this suggestive comment, following which we have stated the loading density of $K_xV_2O_5 \cdot nH_2O$ in the text. In view that the incorporation of $K_xV_2O_5 \cdot nH_2O$ onto NP Au current microcollectors is implemented by an electrochemical process, the loading density of $K_xV_2O_5 \cdot nH_2O$ is estimated to be 1.88 g cm^{-3} on the basis of the passed charge. At the same time, we have recalculated the gravimetric capacity of the constituent $K_xV_2O_5 \cdot nH_2O$ with amorphous, amorphous/crystalline, crystalline structures at various scan rates. As shown in supplementary **Figure S9**, the dual-phase nanostructured $K_xV_2O_5 \cdot nH_2O$ exhibits remarkably enhanced energy storage capability compared with the V_2O_5 -based electrode materials reported previously. As a result, the V-Mn AR-PIMB exhibits energy and power densities superior to aqueous batteries based on Li^+ , Na^+ , K^+ and Zn^{2+} (**Figure 4f**).

(2) It seems the pre-intercalated water plays an important role in determining the optimum structure of the electrode, TGA measurement showing water content will be necessary (only DSC was shown in Figure S3).

Reply: We appreciate the reviewer for this suggestion. Following this suggestion, we have performed TGA measurement of the $K_xV_2O_5$ supported by NP Au. The detailed result is included in supplementary **Figure S3**, demonstrating the change of pre-

intercalated water with the temperature. The process is consistent with the XRD and XPS results.

(3) Authors believed that pre-intercalated hydrated K^+ cations acting as molecular pillars would provide a relative large interlayer spacing for K -ion storage, however, XRD patterns (Figure 2a and S4) didn't suggest the formation of a layered structure.

Reply: We thank the reviewer for this suggestive comment. According to the comment, we have performed additional Raman characterizations of the NP Au/ $K_xV_2O_5$ electrodes during the processes of amorphous/crystalline dual-phase nanostructuring of the constituent ac - $K_xV_2O_5$. As shown in supplementary **Figure S4**, the ac - $K_xV_2O_5$ displays almost the same characteristic Raman bands as the a - $K_xV_2O_5$. Therein, the fingerprints of layer-type structure, such as the skeleton bent vibration at 163 cm^{-1} and the stretching vibration of vanadyl $V=O$ at 1015 cm^{-1} , can be identified to attest that the ac - $K_xV_2O_5$ keeps the layered structure. When further increasing the annealing temperature to $300\text{ }^\circ\text{C}$, it becomes a crystalline mixture composed of orthorhombic V_2O_5 and monoclinic $K_{0.25}V_2O_5$ (m - $K_{0.25}V_2O_5$). Supplementary **Figure S4c** schematically illustrates the atomic structure of the m - $K_{0.25}V_2O_5$, where the $[V_4O_{12}]_n$ sheets consisting of $V(1)O_6$ and $V(2)O_6$ octahedra zigzag chains are linked by oxygen atoms to form 2D layered structure along the (001) plane, with K^+ cations intercalating between the layers. The $[V_4O_{12}]_n$ layers are further connected by $V(3)O_5$ and edge-sharing oxygen atoms to form a 3D tunnel structure. Nevertheless, the c - $K_xV_2O_5$ can be considered as a layered structure in view that the layered o - V_2O_5 is the primary component, as demonstrated by the XRD characterization (**Figure 2a**).

(4) The authors showed a large potential window of 3.2 V (Figure 5d) of the asymmetric cell using a K_2SO_4 electrolyte, it is far beyond the water decomposition potential window of 1.23 V. Will electrolyte (particularly water) decompose to H_2 and O_2 in such conditions?

Reply: In **Figure 5d**, the voltage-time profile corresponds to two AR-PIMBs in series. Therefore, each AR-PIMB device should share 1.6 V, i.e., it stores/delivers charge in a potential window of 1.6 V (**Figure 4b**). Although this potential window is beyond 1.23 V, we do not observe any H_2 or O_2 product during the energy storage/delivery in 0.5 M K_2SO_4 aqueous electrolyte.

Reviewer #2 (Remarks to the Authors)

The referee thinks that the paper should be resubmitted to more specialized journals dealing with battery technology and electrochemistry because of insufficient data and unclear discussion on experimental results.

Reply: We thank the reviewer for his/her comment. According to the reviewers' comments, we have conducted additional experiments and comprehensively revised the manuscript, and feel the paper become much more solid and stronger. In view of the novelty, exceptional performance and scientific implications of the nanoporous metal/oxide hybrid microelectrodes and the microdevices of aqueous rechargeable potassium-ion microbatteries, we wish the reviewer could share our confidence and belief that the work reported in this paper deserves to be published in a high-impact journal, like *Nature Communications*.

Reviewer #3 (Remarks to the Authors)

Li et al. reported a brand-new research strategy for aqueous rechargeable potassium-ion microbatteries (AR-PIMBs). The work innovatively explored the dual-phase $K_xV_2O_5$ nanostructure on Au current microcollectors, which achieved long-term cycling stability. The XRD and TEM prove the dual-phase of $K_xV_2O_5$ nanostructure. On the other hand, the potassium-ion microbatteries constructed with $K_xV_2O_5$ anode and K_xMnO_2 cathode present an excellent performance. The flexible all-solid-state V-Mn AR-PIMBs integrated in solar cells also exhibited an outstanding performance. It is a clear and solid work. I think the paper will lead a new research direction and should be of interest to the readers. I recommend accepting it in Nature Communications after minor revision.

Reply: We thank the reviewer for his/her insightful and constructive comments and finding our work of significant interest. We also appreciate this reviewer for recommending our paper for publication in *Nature Communications*.

(1). The authors believed that Au current microcollectors are important to form dual-phase $K_xV_2O_5$ nanostructure. Thus, how about other current collectors? For example, Cu foil?

Reply: We thank the reviewer for the suggestion. Following this suggestion, we have carried out supplementary experiment to construct microelectrode of $K_xV_2O_5$ on interdigital-patterned nanoporous Cu current microcollectors. However, there takes place a severe corrosion when the $K_xV_2O_5$ electrodeposition is implemented by a potentiodynamic method within a potential window between 0.555 and 1.155 V versus Ag/AgCl in a pH = 1.8 electrolyte. This not only destroys the nanoporous architecture but also fails to electrodeposit the $K_xV_2O_5$ on the Cu current collectors, which are confirmed by SEM and EDS characterizations. These results demonstrate the advantage of nanoporous Au current microcollectors, which are electrochemically stable for the electrodeposition of $K_xV_2O_5$, the subsequent thermal treatments for the formation of dual-phase nanostructure, as well as the flexible microdevices of aqueous rechargeable potassium-ion microbatteries.

(2). In Figure 2, the letter should be rearranged.

Reply: According to this suggestion, we have rearranged the letter in **Figure 2**.

(3). In Figure 3d and Figure S7, is the EIS only straight line? The authors should give more explanation.

Reply: We thank the reviewer for this suggestion. We have give a more detailed description on the EIS results in the main text. From the enlarged EIS spectra, we can find that the EIS spectra in the Nyquist plot are not totally straight lines and there are quasi-semicircles but with small diameters in middle-frequency range. In view that the charge transfer resistance is too small, the EIS looks like a straight line in the full frequency range.

(4). How to increase the value of x of $K_xV_2O_5 \cdot nH_2O$? Is it able to achieve $x = 3$ or higher value?

Reply: We thank the reviewer for this comment. In this work, the constituent $K_xV_2O_5 \cdot nH_2O$ is incorporated into nanoporous gold current microcollectors by an electrodeposition technology in aqueous electrolyte, wherein the x value is controlled by the K^+ concentration from 20 to 60 mM (10 to 30 mM K_2SO_4). It is well-known that nanostructured metals usually are hydrophobic. In order to uniformly incorporate $K_xV_2O_5 \cdot nH_2O$ into the whole nanoporous skeleton, we use a mixture of water and ethanol (3/7 vol/vol) as intermediate to optimize the hydrophilic property of nanoporous gold, and enlist the V and K ions to permeate through skeleton. During the electrodeposition, there forms $K_xV_2O_5 \cdot nH_2O$ to deposit on Au ligaments. However, we only achieve $K_xV_2O_5 \cdot nH_2O$ with the highest x value of 0.25 because of the low solubility limit of K_2SO_4 in the mixture of water and ethanol, as shown in figure shown below.

(5). *Some sentences are too long, and needed to be simplified. For example, in Abstract, line 14-19; Line 86-92*

Reply: According to this suggestion, we have corrected these presentations in the text.

(6). *Some new and important publications in K/Na ion batteries should be referred such as Angew. Chem. Int. Ed. 2018, 57, 8540; Advanced Materials, 2017, 29 (24), 1700989; Advanced Materials, 2017, 29(1), 1604007; Angew. Chem. Int. Ed., 2019, 58, 1484-1488.*

Reply: Following this suggestion, we have added these literatures in the reference list.

Finally, we would like to thank the three reviewers again for their profound help in this work and promoting the scientific significance of this manuscript. In summary we feel the comments/suggestions/corrections by the reviewers to be very insightful and very helpful. Indeed, we feel the paper becomes much more solid and stronger after revision by considering the reviewers' comments.

Reviewers' comments:

Reviewer #1 (Remarks to the Author):

The reviewer recognizes authors had made significant improvement in the revised manuscripts, and largely addressed the questions raised during previous reviewing process. Thus, I recommend the acceptance for the publication in Nat. Common.

Reviewer #4 (Remarks to the Author):

Li et al. report results on the fabricating of dual-phase layered metal oxides as electrodes and their performance in aqueous potassium-ion batteries. The topic of fabricating a high energy density microbattery is interesting and it may be promising in future. However, I may agree with the first two reviewers that ideas as using K⁺ pillared V₂O₅ in aqueous K-ion battery have been seen from the previous work. The novelty is limited. Some comments:

1. The material system of K⁺ pillared V₂O₅ is not of novelty. There are plenty of publications reporting the similar ideas. Why authors chose this material? Please comment on the advantages of K_xV₂O₅.
2. In XRD, peaks of both K_{0.25}V₂O₅ and V₂O₅ appeared, which means authors got a mixture. I am wondering does this mixture affect the electrochemical performance? Or V₂O₅ will be pre-intercalated during the initial cycles? If so, I cannot understand why authors would like to synthesize K_xV₂O₅ first. Please give some comments.
3. In Figure 3a, the CV of K_xV₂O₅ is distorted and has a sharp tail at 0V. What happened at 0V?
4. For all-solid-state device, have authors sealed the devices by, say PDMS? If not, PVA would adsorb water quickly, meaning that this is not solid-state electrolyte.
5. Although the title is "potassium ion microbatteries", the study is more like supercapacitor.

Response to reviewers' comments

Reviewer #1 (Remarks to the Author):

The reviewer recognizes authors had made significant improvement in the revised manuscripts, and largely addressed the questions raised during previous reviewing process. Thus, I recommend the acceptance for the publication in Nat. Commun.

Reply: We appreciate the reviewer for satisfying our revision and recommending our manuscript for publication in *Nature Communications*.

Reviewer #4 (Remarks to the Author):

Li et al. report results on the fabricating of dual-phase layered metal oxides as electrodes and their performance in aqueous potassium-ion batteries. The topic of fabricating a high energy density microbattery is interesting and it may be promising in future. However, I may agree with the first two reviewers that ideas as using K^+ pillared V_2O_5 in aqueous K-ion battery have been seen from the previous work. The novelty is limited. Some comments:

Reply: We appreciate the reviewer for finding that our work of developing high-density aqueous potassium-ion microbattery is an interesting topic and it may be promising in future. We also thank him/her for valuable and insightful comments, which indeed inspire us to highlight the design of electroactive materials, i.e., an amorphous/crystalline dual-phase structure composed of amorphous $K_{0.25}V_2O_5 \cdot nH_2O$ as molecular pillars to support large interlayer spacing and c - V_2O_5 as intermediate to accommodate guest hydrated K^+ cations, in addition to novel and high-performance microdevices and their demo for self-powered systems.

We agree with the reviewer that the materials such as K^+ pillared V_2O_5 have been tentatively used as electrode materials in aqueous potassium-ion batteries in view of vanadium oxides having high theoretical capacity in addition to their advantages such as high earth abundance and low cost. Despite the initial strides that have been made, the superior electrochemical energy-storage performance of layered vanadium oxides has not been fully exploited for practical applications in aqueous batteries, even not to mention rechargeable aqueous potassium-ion microbatteries with high-power and – energy densities. Compared with previous reports on vanadium oxides with/without pre-intercalation of K^+ and their composites with conventional low-dimensional nanostructures (such as nanosheets and nanowires), we would like to emphasize that

our monolithic NP Au/*ac*-K_xV₂O₅ microelectrodes are intrinsically different from all of them in both crystallographic/electrode structures and physical/chemical properties. In previous reports on the K⁺ pillared V₂O₅ materials, they generally have large interlayer distances but are in disordered crystallographic structure due to a large number of water molecules sandwiched between of bilayer V₂O₅ sheets, which essentially block the intercalation/diffusion of guest hydrated cations. At the same time, they also suffer from poor electrical conductivity. Although nanostructured carbon materials (such as carbon nanotubes and graphene) as conductive intermediates are introduced in their composites, the electrodes assembled with these low-dimensional nanostructures via polymer binder not only have poor electrolyte accessibility due to high-density stack but also still exhibit undesirably low conductance because of exceptionally high contact resistances between the composite nanomaterials and the planar current collectors as well as within the composite nanomaterials. In contrast, our monolithic NP Au/*ac*-K_xV₂O₅ microelectrodes have the constituent *ac*-K_xV₂O₅ to be uniformly incorporated into three-dimensional and bicontinuous nanoporous metal current microcollectors via the formation of abundant and stable metal/oxide interfaces. The proper nanoporous microelectrode architecture essentially ameliorates transport kinetics of electrons and ions by offering them pathways along metal ligaments and nanopore channels, respectively. Meanwhile, the abundant metal/oxide interfaces not only facilitate the electron transfer between the electroactive *ac*-K_xV₂O₅ and the current collectors but steadily confine the *ac*-K_xV₂O₅ in the nanopore channels of NP Au for significantly enhanced mechanical/electrochemical durability (**Figure 4e and Figure 5a,b**). Furthermore, in crystallographic structure, the constituent *ac*-K_xV₂O₅ has a unique dual-phase structure composed of amorphous K_{0.25}V₂O₅ *n*H₂O (*a*-K_{0.25}V₂O₅ *n*H₂O) and crystalline V₂O₅ (*c*-V₂O₅). Therein, the *a*-K_{0.25}V₂O₅ *n*H₂O serves as molecular pillars to support large interlayer spacing of the *c*-V₂O₅ and thus offers sufficient room to accommodate more guest hydrated K⁺ cations. This is essentially different from previously reported K⁺ ion pillars in the amorphous V₂O₅, which have to work with water molecules to keep large interlayer distance but lead to poor intercalation/diffusion kinetics of guest hydrated K⁺ ions. As a result, our NP Au/*ac*-K_xV₂O₅ microelectrodes exhibit superior electrochemical energy storage performance, for instance, higher volumetric and gravimetric capacities in the full discharge rates than some of the best electrodes of vanadium oxides with/without the pre-intercalation of K⁺ ions as well as other alkaline ions, as demonstrated in **Figure 3** and Supplementary **Figure 9**.

Besides, we have also completely revised the manuscript according to the reviewer's comments. After revision with these insightful comments constructive suggestions,

we feel the paper becomes much clearer to demonstrate the progress in electrode materials in addition to self-powered microdevices. In view of the novelty of NP Au/ac-K_xV₂O₅ electrode materials, the excellent performances of aqueous rechargeable potassium-ion microbatteries and self-powered microdevices, as well as scientific implications of the nanoporous metal/oxide hybrid microelectrodes, we wish the reviewer could share our confidence together with *Reviewer # 1 (who recommended the acceptance for publication in Nature Communications)* and *Reviewer #3 (who raised only minor issues, as mentioned by Editor)* and belief that the work reported in this paper deserves to be published in a high-impact journal, like *Nature Communications*.

(1) The material system of K⁺ pillared V₂O₅ is not of novelty. There are plenty of publications reporting the similar ideas. Why authors chose this material? Please comment on the advantages of K_xV₂O₅.

Reply: We thank the reviewer for the comment. We agree with the reviewer that recently there are many publications reporting the ideas to make use of alkaline or alkali-earth ions (M = Li⁺, Na⁺, K⁺, Mg²⁺, and Zn²⁺) as molecular pillars to supporting a large basal spacing of vanadium oxides. This is because vanadium oxides have high theoretical capacity in addition to their advantages such as high earth abundance and low cost, and show genuine potential as electroactive materials for aqueous rechargeable batteries. Despite the extensive studies that have been conducted, the superior electrochemical energy-storage performance of layered vanadium oxides has not been fully exploited for practical applications in aqueous batteries, even not to mention rechargeable aqueous potassium-ion microbatteries with high-power and -energy densities. In previous reports on alkaline or alkali-earth vanadium oxides (M_xV₂O₅), they are in either amorphous state filled with a large number of water molecules between bilayer V₂O₅ sheets (*a*-M_xV₂O₅ *n*H₂O) or crystalline monoclinic structure (*m*-M_xV₂O₅) with few water molecules. For the *a*-M_xV₂O₅ *n*H₂O materials, they have large interlayer distances but usually encounter poor intercalation/diffusion kinetics because of water molecules essentially blocking the accommodation of guest hydrated cations during charge/discharge processes. While for the *m*-M_xV₂O₅ materials, although water molecules are squeezed out, their 3D tunnel structures have small tunnel sizes and thus fail to improve the specific capacity and rate capability and alleviate extra massive volume change. In this work, we demonstrate that layered metal oxides, based on a model system of potassium vanadium oxides, with an amorphous/crystalline dual-phase nanostructure, show genuine potential as high-performance electrode materials for aqueous rechargeable in-plane potassium-ion microbatteries when they are integrated on interdigital Au current microcollectors with a three-dimensional and bicontinuous nanoporous architecture,

i.e. NP Au/*ac*-K_xV₂O₅. The constituent *ac*-K_xV₂O₅ is composed of amorphous/crystalline K_{0.25}V₂O₅·*n*H₂O/*o*-V₂O₅ dual phases (**Figure 2c**), wherein the amorphous K_{0.25}V₂O₅·*n*H₂O serves as molecular pillars to keep the large interlayer spacing while the crystalline *o*-V₂O₅ offers sufficient room to accommodate more guest hydrated K⁺ cations via removing secondary-bound interlayer water. As a result, the gravimetric capacity of *ac*-K_xV₂O₅ reaches 382 mAh g⁻¹ at 5 mV s⁻¹ and retains 81 mAh g⁻¹ at 1000 mV s⁻¹, outperforming not only amorphous K_xV₂O₅ (*a*-K_xV₂O₅) and crystalline K_xV₂O₅ (*c*-K_xV₂O₅) (Supplementary **Figure 9a**) but some of the best vanadium oxide-based electrode materials for aqueous energy storage in a full rate range reported previously (Supplementary **Figure 9b**). Associated with the monolithic interdigital Au current microcollectors with a three-dimensional bicontinuous and nanoporous architecture, this enlists the NP Au/*ac*-K_xV₂O₅ electrodes to exhibit a volumetric capacity of as high as ~715 mAh cm⁻³ with an exceptional rate performance.

(2) In XRD, peaks of both K_{0.25}V₂O₅ and V₂O₅ appeared, which means authors got a mixture. I am wondering does this mixture affect the electrochemical performance? Or V₂O₅ will be pre-intercalated during the initial cycles? If so, I cannot understand why authors would like to synthesize K_xV₂O₅ first. Please give some comments.

Reply: We appreciate the reviewer for his/her valuable and insightful comment, in light of which we have realized our *ac*-K_xV₂O₅ electroactive materials are mixtures of *a*-K_{0.25}V₂O₅·*n*H₂O and *c*-V₂O₅ dual phases. Therein, the former serves as molecular pillars to support large interlayer spacing while the latter offers sufficient room to accommodate more guest hydrated K⁺ cations. Owing to the unique dual-phase nanostructure, the gravimetric capacity of *ac*-K_xV₂O₅ reaches 382 mAh g⁻¹ at 5 mV s⁻¹ and retains 81 mAh g⁻¹ at 1000 mV s⁻¹, outperforming not only *a*-K_xV₂O₅ and *c*-K_xV₂O₅ (Supplementary **Figure 9a**) but some of the best vanadium oxide-based electrode materials for aqueous energy storage in a full rate range reported previously (Supplementary **Figure 9b**).

During the initial discharge process, the guest hydrated K⁺ ions will be intercalated into the *c*-V₂O₅ region in the *ac*-K_xV₂O₅. Owing to the support of *a*-K_{0.25}V₂O₅·*n*H₂O pillars, the *c*-V₂O₅ region keeps large interlayer spacing and offers abundant accommodation sites for the guest hydrated K⁺ ions. This unique structure not only facilitate the intercalation/diffusion kinetics of the guest hydrated K⁺ ions but also circumvent the volume change due to the insertion/extraction of the guest hydrated K⁺ ions. This is substantially in contrast with single-phase V₂O₅ with amorphous or crystalline structure, which either suffers from poor intercalation/diffusion kinetics due to the blocking effect of water molecules or encounters extra massive volume

change and thus short lifetime due to narrow interlayer spacing during the insertion/extraction of hydrated K^+ ions. These structural features and electrochemical behaviors are the reasons why we propose a novel amorphous/crystalline dual-phase nanostructure in this work. This facile strategy can be extended to developing other materials systems.

(3) *In Figure 3a, the CV of $K_xV_2O_5$ is distorted and has a sharp tail at 0 V. What happened at 0 V?*

Reply: We appreciate the reviewer for the comment. To demonstrate the reason why the CV of the $ac-K_xV_2O_5$ is distorted and has a sharp tail at 0 V, we have comparatively performed charge/discharge kinetics analysis at the voltage at -0.01 , -0.2 and -0.4 V according to a power-law relationship between the current density (i) and the scan rate (v), namely $i = av^b$, where a is an adjustable parameter, the b value of 0.5 or 1 represents a diffusion- or surface-controlled process, respectively. The plot of $\ln(i)$ vs $\ln(v)$ is shown in Supplementary **Figure 8b**. As we can see, the b value at -0.01 V is 0.77 in the full range of scan rate, implying that there always takes place electrochemical extraction of hydrated K^+ ions. While at -0.2 and -0.4 V with the scan rates below 100 mV s^{-1} , the NP Au/ $ac-K_xV_2O_5$ only exhibits a surface-controlled process. Therefore, we can observe a sharp tail at 0 V in the CV of the $ac-K_xV_2O_5$ at 50 mV s^{-1} , as shown in **Figure 3a**.

(4) *For all-solid-state device, have authors sealed the devices by, say PDMS? If not, PVA would adsorb water quickly, meaning that this is not solid-state electrolyte.*

Reply: We thank the reviewer for the comment. We have used PDMS to seal the microdevices and mentioned it in Method section.

(5) *Although the title is “potassium ion microbatteries”, the study is more like supercapacitor.*

Reply: Considering that aqueous electrolyte usually has ionic conductivity two orders of magnitude higher than organic electrolytes, in this work, we focus on developing aqueous rechargeable microdevices that can store/deliver energy with lithium-ion microbattery-like capacity and microsupercapacitor-like rate performance. With this aim, we construct potassium-ion microbatteries by integrating $ac-K_xV_2O_5$ anode and K_xMnO_2 cathode on interdigital nanoporous Au current microcollectors (V-Mn AR-PMIBs) and investigate their electrochemical performance at high charge/discharge rates. As a result of unique electrode structure, in which the constituent $ac-K_xV_2O_5$ has large interlayer spacing to accommodate guest hydrated K^+ ions and the NP Au skeleton facilitates electron and ion transports, the NP Au/ $ac-K_xV_2O_5$ electrodes exhibit a volumetric capacity of as high as $\sim 715 \text{ mAh cm}^{-3}$

with an exceptional rate performance. This enlists the AR-PIMBs to store/deliver charge with volumetric energy of $\sim 103 \text{ mWh cm}^{-3}$ at electrical power comparable to carbon-based microsupercapacitors. The volumetric energy is about fourteen-fold and seventy-fold higher than that of 4 V-500 μAh thin-film lithium battery and onion-like carbon microsupercapacitors, respectively. The superior electrochemical properties make them promising candidates as micropower sources to complement other energy conversion devices such as commercial solar cells.

REVIEWERS' COMMENTS:

Reviewer #4 (Remarks to the Author):

The authors have addressed most of my concerns and the manuscript has been improved. I can give a green light.

Response To Reviewer's Comments

Reviewer #4 (Remarks to the Author):

The authors have addressed most of my concerns and the manuscript has been improved. I can give a green light.

Reply: We appreciate the reviewer for his/her positive comments and recommendation for publication in *Nature Communications*.